# How do care home staff use non-pharmacological strategies to manage sleep disturbances in residents with dementia: The SIESTA qualitative study

Lucy Webster [1,2], Sergi G. Costafreda [1,3], Kingsley Powell [1], Gill Livingston [1,3]*

1 Division of Psychiatry, UCL, London, United Kingdom, 2 Centre for Health Services Studies, University of Kent, Canterbury, United Kingdom, 3 Camden and Islington NHS Foundation Trust, London, United Kingdom

* G.livingston@ucl.ac.uk

**Data Availability Statement:** We have now added the data to a public repository and it is accessible on the link here: https://reshare.ukdataservice.ac.

## Abstract

### Background

Sleep disturbances affect 38% of care home residents living with dementia. They are often treated with medication, but non-pharmacological interventions may be safer and effective yet more difficult to implement. In the SIESTA study (Sleep problems In dEmentia: interviews with care home STAff) we explored care home staffs' experience of managing sleep disturbances in their residents living with dementia.

### Methods

We conducted one-to-one semi-structured interviews in four UK care homes, and purposively recruited a maximum variation sample of 18 nurses and care assistants, who were each interviewed once. We used a topic guide and audio-recorded the interviews. Two researchers independently analysed themes from transcribed interviews.

### Results

Staff used a range of techniques that often worked in improving or preventing residents' sleep disturbance. During the daytime, staff encouraged residents to eat well, and be physically active and stimulated to limit daytime sleep. In the evening, staff settled residents into dark, quiet, comfortable bedrooms often after a snack. When residents woke at night, they gave them caffeinated tea or food, considered possible pain and discomfort, and reassured residents they were safe. If residents remained unsettled, staff would engage them in activities. They used telecare to monitor night-time risk. Staff found minimising daytime napping difficult, described insufficient staffing at night to attend to reorient and guide awake residents and said residents frequently did not know it was night-time.

### Conclusions

Some common techniques, such as caffeinated drinks, may be counterproductive. Future non-pharmacological interventions should consider practical difficulties staff face in

uk/855437/ with the DOI: 10.5255/UKDA-SN-855437.

**Funding:** The SEISTA study was funded as part of an Economic and Social Research Council (ESRC) funded PhD studentship. The funders had no role in study design, data collection and analysis, decision to publish or preparation of the manuscript.

**Competing interests:** LW reports that the analyses in this paper were undertaken as part of their ESRC funded PhD studentship. SGC reports receiving research funding from Alzheimer's Research UK Grant, European Research Council and Dunhill Medical Trust. GL reports participating on the Lancet Healthy Longevity Board and receiving research funding from Alzheimer's Society, Wellcome trust, and NIHR. This does not alter our adherence to PLOS ONE policies on sharing data and materials.

managing sleep disturbances, including struggling to limit daytime napping, identifying residents' night-time needs, day-night disorientation, and insufficient night-time staffing.

# Introduction

Sleep disturbances affect 38% of people with dementia living in care homes [1], and commonly include awakening at night, getting up at night, and excessive daytime sleepiness [2]. Sleep disturbances can negatively impact the residents who experience them, other residents who are disturbed, and care home staff [1, 3, 4]. They are therefore a priority for treatment [5], but there is currently a lack of evidence for potential treatments [6]. Hypnotic medications are a class of medication commonly used to manage sleep disturbances, and there are no randomised controlled trials of hypnotics in people living with dementia so their efficacy is unclear [7]. In people living with dementia and older adults residing in care homes, hypnotic medications are associated with an increased risk of falls and mortality [8–10], and in the United Kingdom (UK) hypnotics are not recommended by National Institute for Health and Care Excellence (NICE) as first-line treatments in older adults or people living with dementia [11, 12].

There is no conclusive research into how sleep disturbances can be treated non-pharmacologically in people with dementia. As the causes of sleep disturbances in dementia are often complex, the best treatments are thought to be multi-component interventions [13]. The most recent NICE guidelines recommend personalised multi-component approaches to sleep disturbances, which include sleep hygiene, increasing exposure to daylight, and increasing exercise and personalised activities [12]. Recent pilot studies provide preliminary evidence of the efficacy of multi-component interventions in community settings [12, 14–16].

There are additional factors in care homes, compared to the community, that may cause sleep disturbances in residents with dementia. These factors include noise [17], lights being on [18], other residents' wakefulness and carers conducting night-time incontinence care or observations of residents for their safety [19]. Care home residents may also spend a long time in bed at night due to the routines within the homes [20, 21] which can fragment and disturb sleep [22]. Therefore successful management and implementation of interventions for residents' sleep disturbances in care homes require consideration of care home organisation and environment [23]. A recent systematic review of multi-component non-pharmacological interventions for sleep disturbances in nursing home residents found none of the studies reported considering barriers to implementation in care homes [24].

Here we report results from the SIESTA (Sleep problems In dEmentia: interviews with care home STAff) study, where we explored the views and experiences of care home staff members in one-to-one semi-structured interviews on how sleep disturbances in residents with dementia are managed non-pharmacologically in practice and any barriers to implementing these strategies.

# Methods

## Setting and sampling

The SIESTA study was approved by UCL Research Ethics Committee (number 14289/001). We have previously published results from the SIESTA study on the negative impacts sleep disturbances can have on residents with dementia [3]. We recruited four care homes in Greater London, United Kingdom, and interviewed 18 care home staff members. Eligible staff were

either care home nurses or care assistants who currently or within the previous year had worked a mix of both day and night shifts.

We purposively recruited the four care homes to ensure a mix of homes providing nursing or residential care, and not-for-profit or privately funded. The definition of residential care homes in the UK is *"care homes which only provide accommodation and personal care"* whereas nursing homes are defined as *"care homes which provide personal care and nursing"* [25]. The homes were in both urban and suburban areas, owned by different providers, with home sizes ranging from 48–215 beds. The latest statutory inspections by the UK Care Quality Commission rated three of the care homes as good and one as outstanding. We purposively recruited a maximum variation sample of care home staff members, interviewing nurses and care assistants who were diverse in terms of role, age, sex, and ethnicity. We continued until we reached theoretical saturation when any additional interviews did not add new information to the data already collected [26].

## Procedure

We contacted the managers of eligible care homes to ask for permission to recruit their staff. Participating care home staff then gave written informed consent before the interview and also completed a demographic questionnaire including their role in the care home, how long they have worked in the present care home and care homes in general, their shift pattern, sex, age, and ethnicity. We did not consent the individual residents that staff spoke about during the interviews as we were asking staff about their experiences of caring for these residents, and we also asked them not to tell us any residents' names or personal identifiable details. This approach was approved by the UCL Research Ethics Committee.

All interviews were conducted by LW in a private room of the care home and audio recorded. Interviews lasted for 20–60 minutes, with an average of 40 minutes. Interviews were one-to-one and semi-structured, using a topic guide to explore staff's experiences of sleep disturbance in residents with dementia, including the management. We developed the topic guide (S1 File) from the literature including our systematic review [1] and altered the guide iteratively, as and when new topics were discussed in the interviews. Care home staff were asked to focus on three residents who they knew well who had dementia and sleep problems, with each resident discussed separately. We asked those who at the time of the interview were working solely on day or night shifts to consider residents that they knew well from when they had worked mixed day and night shifts. As a token for the time taken each participant received a £20 voucher.

## Data analysis

Interviews were transcribed verbatim by an external transcription service and anonymised. We used NVivo11 to manage the data and used reflexive thematic analysis which is *"a method for systematically identifying, organizing, and offering insight into patterns of meaning (themes) across a data set"* [27]. We decided that thematic analysis was the most appropriate analysis method because we wanted to understand the experiences and thoughts of the care home staff members across the dataset, searching for common and shared meanings of the causes and impacts of sleep disturbances, with less concern for unique or individuals experiences or meanings [28, 29].

We followed the six phases described by Braun and Clarke [30]: familiarising yourself with your data, generating initial codes, searching for themes, reviewing themes, defining and naming themes, and producing the report.

We developed a thematic coding framework, using the data from the first five interviews were initially independently coded by authors LW and KP. They identified the main themes and then developed an initial coding framework in line with the study objectives. LW and KP then met and discussed and edited this framework together. The initial coding framework was then discussed and agreed upon by all study authors. Each additional interview was coded independently by LW and KP using the framework, which was added to iteratively. They met after every 3–4 interviews were coded, discussed any coding disagreements and a consensus was reached. If needed a third person (GL) was consulted for any coding disagreements where a consensus was not reached.

## Results

### Participant demographics

We interviewed 18 staff members working in four care homes in Greater London; two were privately funded and two not-for-profit, one of which was a residential home and three homes provided nursing and residential care.

Participants were interviewed between November 2018-May 2019 and discussed approximately 54 different residents living with dementia. Nine participants worked in privately funded homes and nine in not-for-profit homes. The staff interviewed were mostly female (see demographics in Table 1), with five working in residential care homes, and 13 working in care homes providing nursing and residential care. Staff varied in terms of their ethnicity and their roles within the care homes, and their age ranged from 21–64 years. They had worked in care homes for between 1–25 years, with a median of six years of care home experience. Fourteen staff members were interviewed during their day shift, two were interviewed whilst working a night shift, and the last two were interviewed before and after working a night shift, respectively.

**Table 1. Sociodemographic and work-related characteristics of staff participating in the SIESTA study.**

| Characteristics | | Participants |
|---:|:---:|---:|
| **Sex** | Male | 2 |
| | Female | 16 |
| **Age (years)** | Range | 21–64 |
| | Median | 38 |
| **Ethnicity** | Black British African | 7 |
| | Black British Caribbean | 2 |
| | Filipino | 3 |
| | Mauritian | 2 |
| | South American | 1 |
| | White British | 1 |
| | White Other | 2 |
| **Time working in the current care home** | Range | 1–15 years |
| | Median | 5 years |
| **Time working in care homes** | Range | 1–25 years |
| | Median | 6 years |
| **Role in the care home** | Care assistant | 8 |
| | Senior care assistant | 6 |
| | Team leader | 2 |
| | Nurses | 2 |

## Management of sleep disturbances

The management of sleep disturbances in residents with dementia ranged from prevention strategies during the daytime, to evening strategies to promote sleep, and strategies in the night-time to manage when a resident was up and awake. Staff also described common barriers to managing sleep disturbances in their residents with dementia.

## Daytime strategies to prevent night-time sleep disturbances

To prevent night-time sleep disturbances, staff tried to ensure that residents ate well during the daytime so that they would be less likely to wake due to hunger during the night. Staff also tried to keep residents active and socially stimulated during the day to promote better night-time sleep.

> *The night staff, they appreciate it when we keep him up and give him drinks and snacks rather than them having to get him up and then put him back to bed... He sleeps more in the night-time. (9; female care assistant)*

> *I'd say now he sleeps really well because he's active during the day. We're walking him a lot so he's getting a lot of exercise. (9; female care assistant)*

Activity in the daytime was also used to encourage residents to stay awake, and staff tried to reduce residents napping where possible.

> *And we also tried to lessen the sleep they have during the day... Like, say maybe two hours. (7; female nurse)*

> *When they don't sleep well at night... to put them in bed, even for two hours to get a little rest after their meal. And then we can take them back out and have some form of activity with them. (13; female senior care assistant)*

## Evening strategies to promote sleep

In the evening, staff would promote good sleep hygiene, such as using wind-down strategies, ensuring bedrooms were quiet and dark. They would also sometimes give residents food or a warm drink just before bedtime, as they had noticed residents may then fall asleep more easily.

> *We try to be quiet, to make him quiet, like switch off all the lights, like no doors open, windows closed, he likes in dark. (1; female care assistant)*

> *We've noticed that if you give someone food and they'll feel full and satisfied so they can just easily go to sleep. Or sometimes warm milk. (7; female nurse)*

They sometimes showed residents that it was night-time if it was dark outside. They would ensure that residents felt comfortable and settled before they were left alone in their bedrooms to fall asleep.

> *We always try and tell and show him the time and say, "it's night-time." We draw the curtain and say, "look it's pitch dark outside." (3; female team leader)*

> *We give them a drink every night before they go to bed... we wash them and we settle them. (12; female care assistant)*

### Night-time strategies to manage sleep disturbances

**Food and drink.**  Food was often given to residents who were awake during the night-time, as staff thought that residents may be awake due to hunger, particularly for residents who did not eat well during the daytime because they were napping. A cup of tea was also commonly given to residents in the night-time, and mostly this was caffeinated tea, rather than decaffeinated.

> *That's why at night. . . we always have a lot of sandwich[es]. Every time he pass[es], just go, and then he'll grab one. Because during the day, he doesn't eat much. (15; female care assistant)*

> *Yes, we just kind of say, "we're going to get you a cup of tea. I know you'll have a cup of tea, you would love that." And then he say[s], "yes, yes, I would love that." (6; female senior care assistant)*

Giving residents food and tea was sometimes perceived as counterproductive by our interviewees as reinforcing a disturbed sleep pattern. Staff discussed how residents would wake in the night-time as they had become used to eating and drinking more frequently at night, and then consequently eat little during the day and again be hungry at night.

> *She's awake in the night, asking for a cup of tea, asking for biscuits, asking for food. (11; female senior care assistant)*

**Comfort and company.**  Comfort and a cup of tea were often used together, with staff using the time that the resident was drinking their tea as an opportunity to provide them with companionship during the night-time. Staff would reassure residents who awoke in the night that staff were there for company and alleviate any concerns. When residents were awake and up during the night, staff would also reorient them by telling them that it was night-time, and time to sleep.

> *You come and chat with the person, at night, having a cup of tea together. . . it's all like building a relationship. (14; male senior care assistant)*

> *I said, "everything is fine. Your family are fine, you don't need to worry." So, we do look into techniques. . . to reassure him, to encourage him to go to sleep. (6; female senior care assistant)*

**Occupying and tiring the resident.**  Additionally, when residents got up during the night and went outside of their rooms and into the common spaces of their homes, staff would try to guide them back to their bedrooms and into their beds. When staff felt that residents would not be able to go back to sleep straight away, and may become agitated, they went along with what the person wanted to do or an activity the resident enjoyed, for example, playing card games or golf.

> *Sometimes you have to encourage her to go back to sleep. She'll tell you "No, I'm fine here." You take her back and she will come back. (5; female care assistant)*

> *He likes to play golf. Sometimes during the night, you find us playing golf with him because he's not sleeping. (15; female care assistant)*

Staff would also walk around the home or unit with residents who were up and agitated or seemed to have lots of energy, in the hopes that it would make them feel tired again and to help manage their agitation. For some residents who became agitated in the night, staff would work with their families to consider strategies to manage this. They would also sometimes give misinformation to the resident in order to reassure and comfort them when they were agitated in the night.

*She's getting up all the time, throwing the legs over, so the best thing to do sometimes when she's really agitated and unsettled is to take her for a walk around the unit so she can get a little bit more exhausted. (12; female care assistant)*

*We found that for him, it works better to say, "the taxi's coming tomorrow morning, or your wife's coming tomorrow morning with the dog, and you guys are going to go home together because the cars at the mechanic". Because that's what his wife told us to say. (15; female care assistant)*

**Assessing and manage pain and discomfort.** All staff interviewed discussed how it was common for residents to be in pain during the night or to be experiencing discomfort, for example from needing to use the toilet, and would monitor residents for this and manage residents' pain and discomfort.

*If she will keep saying that she's got sharp pain, we will call the nurse because they know better how to reassure them and if they need any medication or the painkillers. (10; female senior care assistant)*

*I always tell the carers that, including myself, if because of wandering about, can we just direct him to the toilet? And then, see. . . So, most of the time also, it will solve the problem. (17; male senior care assistant)*

## Monitoring residents

Staff would also monitor if residents were awake with regular, often hourly, checks and observations. For some residents who were felt to be at risk if they walked around the home at night, staff had a plan in place where a member of staff sat outside or near the resident's room and therefore would quickly become aware if the resident was up.

*So, we just keep on checking if she went back to sleep, or she's still. . . Oh, she's awake for how many hours? Oh, she went back to sleep. (4; female care assistant)*

*No, because we have a carer that will sit outside her door and waits in the corridor. She will come to you, first of all, so she doesn't wander. (9; female care assistant)*

Sometimes telecare, in particular sensor mats on the bed or next to the bed on the floor, were used to monitor when residents were getting up, particularly for residents at risk of falls or for those who commonly disturbed other residents.

*So, we always had a bed sensor around. So as soon as he moves, the bed sensor is triggered. So, when you see him. . . he's already sitting on the edge of the bed by the time you reach his room. (6; female senior care assistant)*

*He did manage to get into one of the resident's rooms at night-time, which frightened the person. But now. . . we put [in] a sensor mat. So, whenever he gets up it alerts the team. (17; male senior care assistant)*

## Barriers to managing sleep disturbances

There were often barriers to using strategies to manage sleep disturbances. Staff would know that residents needed to stay awake in the day to sleep well at night but often found it difficult to encourage and keep people awake in the daytime, even with activities and social stimulation.

*We try to keep him awake but it's no chance. You can't. He just sit[s] down and he's too tired and he close his eyes. (1; female care assistant)*

*Earlier on this month there was a time he was so tired he would just eat a piece of his breakfast and go to bed and sleep. We try as much to keep him busy during the day but he's quite a strong headed person. (3; female team leader)*

Staff described finding it difficult during the night to encourage residents to go back to their rooms and to bed to try to sleep. Staff sometimes also found it difficult to understand what residents needed or wanted during the night-time if residents were unable to communicate this to staff.

*If he doesn't want to go to his room, we just have to leave him because we just don't want him to kick you or hit you. So, whenever he's sleeping in the chair, and you try maybe, "go to your room", one, two, three [times]. He says, "no." (5; female care assistant)*

*Because she couldn't really express every time that this is what I want. . . So, they don't know what causes the agitation at night. (7; female nurse)*

Due to their dementia, some residents did not understand why staff were trying strategies or why they wanted them to sleep, as they did not recognise that it was night-time.

*So, it's difficult in a sense that he can't really understand why we want him to stay in bed. (7; female nurse)*

*Especially when the wintertime and the nights are very long, so he wake[s] up and he don't know what he's doing. He's saying that he's going for supper. . . it's morning. (1; female care assistant)*

Staff also described understaffing at night-time as a barrier to providing the time and care needed to help residents manage their sleep disturbance, which led to some staff feeling guilty.

*I think it's just loneliness because sometimes we'll sit with her, and we can chat for hours. Sometimes we don't have the time. (15; female care assistant)*

*Although we have some nights that we're short. . . Here in this floor, it should be four staff working. One is one to one, two staff on the floor and then the nurse. So, if one has cancelled their shift so the nurse will be working on the floor or sometimes sitting as one to one and then the other two staff will be ensuring the safety of other residents. (7; female nurse)*

Occasionally for some residents, the staff we interviewed perceived that the residents families objected to staff putting potentially effective strategies into place, which was perceived as a barrier.

*Because the son is always complaining, "my mum is not comfortable. She has to go to bed." So sometimes exactly after lunch, he'll want his mum to go to bed. (5; female care assistant)*

*So, the nurse told us that we should reduce the wine in the daytime. Because it's making him sleep less in the night. But when his children comes [to visit], her daughter, when she comes, she brings some from the house. (16; female care assistant)*

## Discussion

Staff used a range of strategies to manage sleep disturbances. Firstly, they would try to prevent night-time sleep disturbances by ensuring that in the daytime residents ate well and were active and napped as little as possible. Strategies to promote good sleep hygiene included ensuring a dark and quiet environment conducive to sleep, ensuring that residents were not hungry before bed, and felt comfortable and settled [12].

Strategies used in the night-time also included comforting residents, encouraging residents to go back to their bedrooms or sleep, comforting and reassuring residents, considering and managing pain and discomfort and giving residents food. While staff also described cups of tea as a management tool to comfort residents, caffeinated drinks are advised against due to the counter-productive effects of caffeine on sleep [31]. Increased fluid intake immediately before bedtime could also increase the chances of resident's night-time awakening to use the toilet. Staff described barriers to managing sleep disturbances such as getting residents to stay awake and minimising napping during the day. During the night-time, staff reported finding it difficult to guide residents back to their rooms, to help residents understand it was night-time, and to be able to accurately know what the individual with dementia needed or wanted if they were unable to communicate this to care home staff.

### Comparison to previous literature

A previous qualitative study on sleep disturbances in care homes interviewed family carers and care home staff and found that both family carers and staff described pharmacological interventions as predominant treatment strategies, with staff also describing fewer examples of non-pharmacological strategies than family carers did [32]. In comparison, in our study care home staff often described pharmacological strategies as last resort for residents who had severe sleep disturbances that had not improved with non-pharmacological strategies and discussed side effects [3]. The main examples provided by staff in the previous study [32] were social stimulation and offering food and drink, but the staff interviewed did not describe these as treatments for sleep disturbances as such.

Some of the non-pharmacological strategies staff in the SIESTA study described for managing sleep disturbances may also be counterproductive, such as encouraging prolonged napping in the day if someone slept poorly the night before. Furthermore, giving people food during the night might reinforce their sleep disturbance, as they may associate nighttime with feeding time, potentially maintaining the cycle of sleep disturbances [33]. Also, residents were often given caffeinated tea at night, which could then keep residents awake or wake them up later [34]. None of our informants mentioned early bedtime as a problem, despite it being discussed as a systemic barrier to good sleep [21]. Daytime light therapy has mixed evidence to improve night-time sleep disturbances [35, 36], however, staff did not mention it as a strategy, possibly because they are not aware of its use.

Staff described a range of barriers to implementing strategies to reduce sleep disturbances. For example, staff found it difficult to keep residents awake in the daytime, making it difficult to encourage residents to be more physically and mentally active and to nap less during the daytime. People with dementia living in care homes are often not very active [37, 38] and may spend a large proportion of the daytime sitting down [39]. This could increase the chances of

residents napping in the day, as well as residents being inactive and missing out on daytime activities, all of which could negatively impact sleep [13]. Furthermore, the hypnotic medications residents are given for their sleep disturbances could also make them fatigued and sleepy during the day [40], leading to them needing to nap excessively during the daytime and possibly exacerbating residents sleep disturbances [3].

A staff-related barrier raised by participants was staffing levels on night shifts when fewer staff are working, as it is assumed that residents are sleeping so there is a lower need for staff, which may not always be true [41]. This was also mentioned in the previous qualitative study [32] when they conducted focus groups with family carers, but care home staff who were informally interviewed had mixed views on whether this was an issue.

In the current COVID-19 pandemic, due to care home restrictions, residents with dementia experienced changes in the home environment and their day-to-day life which may have impacted their sleep [23] including a reduction in activity and social interaction as they are often prevented from mixing. For example, for infection control, many care homes have stopped or limited family visits, group activities, and communal dining, and residents are also reported to have spent a lot more time alone in their bedrooms [42]. Therefore, the effective management of sleep disturbances may be of even greater importance, but also the pandemic may have created further barriers to management.

A strategy that some of our staff interviewees reported was providing misleading information to residents to encourage sleep, such as reassurance by telling them they are going home the next day, which was a strategy sometimes supported by families in our sample. This is ethically problematic and could also potentially lead to a lack of trust if residents remember the misinformation. Furthermore, some of the strategies staff use to promote wakefulness in residents in the daytime could also be perceived as potentially coercive, for example, residents have preferences if they want to participate in activities or nap instead, but there are often discrepancies between care home routines and residents' preferences [43].

It would also be beneficial to use the results of this study when considering the development and implementation of future non-pharmacological interventions. These future non-pharmacological interventions are likely to be multi-component, and could include trying to avoid residents having early bedtimes, as many care home residents often do due to the care routines within the homes [20, 21, 44], and spending an extended time in bed is associated with worse sleep [22]. Interventions could also consider ensuring activities are provided just after lunch when residents are most likely to engage in daytime sleep. Offers of food and drink at night could be gradually phased out and replaced by sufficient nutrition and hydration during the day, to reduce the incentives for residents to wake up at night. Assessment and management of pain should also be considered, ensuring that staff are skilled in assessing both acute and chronic pain and that protocols are in place for different ways to manage residents' pain as undertreated pain is common in care home residents and can affect sleep [45–47].

## Strengths and limitations

We interviewed staff who worked a mix of day and night shifts, which was a strength of our study because both groups of staff offer complementary perspectives, from staff who see the residents in the night-time when they would normally be expected to be sleeping, as well as from those who see the potential effects of sleep disturbance during the daytime. We also recruited a maximum variation sample to ensure we included staff with a wide range of demographic characteristics. It reflected the composition of staff teams who work in care homes, as the majority of care home staff are female [48] and from a mix of ethnicities, including many who speak English as an acquired language [49]. However, our sample may be less reflective of

the staff who work in care homes in less diverse areas than Greater London, and therefore the results may be less generalisable.

We recruited staff from a mix of residential and nursing homes to ensure we had a range of experiences. Three of the four care homes were rated as "good" by the Care Quality Commission (CQC), the government regulator in England which reviews health and social care services, and rates on a four-level scale from outstanding to inadequate in their CQC ratings, and one was rated as outstanding. This may be better than average for care homes across England, 84% of which were rated as good or outstanding by the CQC in 2019 [50]. As the care homes chose to participate in the research, had participated in research previously, and had high CQC ratings which usually indicated higher quality homes, and this may introduce selection bias [49], and limit the results of the study's generalisability to care homes of lower quality.

We developed a coding framework of themes using data from the first five interviews and added to this and changed it iteratively as new themes were developed from the data. By asking staff to choose three particular residents who had sleep disturbances, we focused on real practical experiences rather than hypothetical ones. The staff were interviewed about their experiences, which even though individual to them, are influenced by the environment and organisational structure within which they work. The data could have been impacted by organisational factors in terms of staff hierarchies [51], prioritisation of care [52], staffing levels [53], and the care home provider, all of which could influence the management of sleep disturbances. In the commentary written about our previous paper from the SIESTA study on the impact of sleep disturbances [5], the authors comment that staff may be less likely to discuss residents who are awake but lying still and quiet, as they may miss that these residents are awake during their hourly observations. Furthermore, they also comment that the SIESTA study would have been strengthened by measuring sleep disturbances in the residents to see if staff were more likely to discuss those who were awake and disruptive than those who lay awake still and quietly. We acknowledge these as limitations, inherent in the fact that without direct monitoring of sleep, residents who remain in their bed quiet are less likely to come to the attention of carers, and therefore they were less likely to report on them for our study. Furthermore, staff were asked to discuss their experiences of caring for residents with dementia and did not check the diagnoses of the residents, though many people living with dementia in care homes will not have a formal diagnosis [54].

## Conclusions

With sleep disturbances often having a negative impact on residents, and as there are currently no safe, efficacious and implementable treatments, future research needs to focus on developing and implementing suitable and efficacious nonpharmacological treatments [55]. Appropriate management strategies may also help reduce the distress these disturbances can cause for staff themselves, and other residents, reducing the broader impact [4, 5]. In the daytime, ways to reduce or stop napping would also be an important element for the management of sleep disturbances, by encouraging residents to do more activity and feel mentally stimulated during times of the day when they are more likely to nap, for example in the afternoon after eating lunch. It would also be important to consider the difficulty staff described to keep residents awake in the daytime in the SIESTA study, as well as other barriers staff described like residents not being able to understand why the staff member is trying to help them back to their room to sleep and staffing levels on night shifts.

## Supporting information

**S1 File. Qualitative interview topic guide.**
(TIF)

## Acknowledgments

We thank all of the care homes and staff who participated in the SIESTA study.

## Author Contributions

**Conceptualization:** Lucy Webster, Sergi G. Costafreda, Gill Livingston.

**Data curation:** Lucy Webster, Sergi G. Costafreda, Kingsley Powell, Gill Livingston.

**Formal analysis:** Lucy Webster, Kingsley Powell.

**Funding acquisition:** Lucy Webster, Sergi G. Costafreda, Gill Livingston.

**Investigation:** Lucy Webster.

**Methodology:** Lucy Webster, Sergi G. Costafreda, Gill Livingston.

**Project administration:** Lucy Webster.

**Supervision:** Sergi G. Costafreda, Gill Livingston.

**Writing – original draft:** Lucy Webster.

**Writing – review & editing:** Lucy Webster, Sergi G. Costafreda, Kingsley Powell, Gill Livingston.

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
