## [Decision Letter · Decision Letter 0]

10 Jan 2022

PONE-D-21-30400Describing and overcoming barriers to care home staff using non-pharmacological strategies for sleep disturbances in care home residents with dementia: the SIESTA qualitative studyPLOS ONE

Dear Dr. Webster,

Thank you for submitting your manuscript to PLOS ONE. After careful consideration, we feel that it has merit but does not fully meet PLOS ONE’s publication criteria as it currently stands. Therefore, we invite you to submit a revised version of the manuscript that addresses the points raised during the review process.

All the three reviewers have highlighted positive aspects in the manuscript. Yet, they all have comments at the methodological and conceptual level that I kindly ask the authors to carefully address. On the basis of the revision I will be able to make an evaluation about publication. 

We look forward to receiving your revised manuscript.

Kind regards,

Sara Rubinelli

Academic Editor

PLOS ONE

Journal Requirements:

2. Please describe in your methods section how capacity to provide consent was determined for the participants in this study. Please also state whether your ethics committee or IRB approved this consent procedure. If you did not assess capacity to consent please briefly outline why this was not necessary in this case.

The SEISTA study was funded as part of an Economic and Social Research Council (ESRC) PhD studentship awarded to LW, and surpvised by SC and GL. 

4. Thank you for stating the following in the Acknowledgments/ Funding Section of your manuscript: 

The SEISTA study was funded as part of an Economic and Social Research Council (ESRC) funded PhD studentship. The funders had no role in study design, data collection and analysis, decision to publish or preparation of the manuscript.

The SEISTA study was funded as part of an Economic and Social Research Council (ESRC) PhD studentship awarded to LW, and surpvised by SC and GL. 

LW reports that the analyses in this paper were undertaken as part of their ESRC funded PhD studentship. SGC reports receiving research funding from Alzheimer's Research UK Grant, European Research Council and Dunhill Medical Trust. GL reports participating on the Lancet Healthy Longevity Board and receiving research funding from Alzheimer's Society, Wellcome trust, and NIHR. 

6. Please note that in order to use the direct billing option the corresponding author must be affiliated with the chosen institute. Please either amend your manuscript to change the affiliation or corresponding author, or email us at plosone@plos.org with a request to remove this option.

7. We note that you have stated that you will provide repository information for your data at acceptance. Should your manuscript be accepted for publication, we will hold it until you provide the relevant accession numbers or DOIs necessary to access your data. If you wish to make changes to your Data Availability statement, please describe these changes in your cover letter and we will update your Data Availability statement to reflect the information you provide.

8. PLOS requires an ORCID iD for the corresponding author in Editorial Manager on papers submitted after December 6th, 2016. Please ensure that you have an ORCID iD and that it is validated in Editorial Manager. To do this, go to ‘Update my Information’ (in the upper left-hand corner of the main menu), and click on the Fetch/Validate link next to the ORCID field. This will take you to the ORCID site and allow you to create a new iD or authenticate a pre-existing iD in Editorial Manager. Please see the following video for instructions on linking an ORCID iD to your Editorial Manager account: https://www.youtube.com/watch?v=_xcclfuvtxQ

Reviewers' comments:

Reviewer's Responses to Questions

**Comments to the Author**

1. Is the manuscript technically sound, and do the data support the conclusions?

Reviewer #1: Yes

Reviewer #2: Yes

Reviewer #3: Partly

2. Has the statistical analysis been performed appropriately and rigorously? 

Reviewer #1: Yes

Reviewer #2: N/A

Reviewer #3: No

3. Have the authors made all data underlying the findings in their manuscript fully available?

Reviewer #1: No

Reviewer #2: Yes

Reviewer #3: Yes

4. Is the manuscript presented in an intelligible fashion and written in standard English?

Reviewer #1: Yes

Reviewer #2: Yes

Reviewer #3: Yes

5. Review Comments to the Author

Reviewer #1: Staff used a range of techniques that often worked in reducing or preventing sleep

disturbance. Some common techniques, such as caffeinated drinks, may be

counterproductive. Non-pharmacological interventions should consider practical

difficulties in limiting daytime napping, identifying residents night-time needs; day-night

disorientation, and insufficient night staffing. This is an useful study. I find appropriate a short introduction in aging and sheep disturbances, useful.

Reviewer #2: This is an interesting manuscript that reports a qualitative study on factors that affect sleep behaviour for people with dementia in residential care settings. An adequate number of interviews were undertaken across four facilities, and the results are presented in a well-structured form. Overall, an important contribution.

I would consider the manuscript to be very well constructed and have no particular suggestions regarding the methodology and results. However, the paper does bring to light certain practises and attitudes that perhaps could be further pursued or at least discussed. In such a qualitative study, it would have been useful to understand the care ‘lens’ through which aged care workers and nurses undertook their roles. While it is laudable that staff would pursue non-pharmacological means to reduce sleep disturbances, from the descriptions of attitudes, it was not clear how person-centred these approaches may be. While intentions may have been good, many of the described practises could be perceived as manipulative and coercive, especially around activities and attempts to promote wakefulness during the day. I am wondering if the authors may have data on attitudes regarding the autonomy and ability for residents to make choices about their daily and nightly activities? From some descriptions, certain practices could be interpreted as physical restraint. It may not be available, but it would be interesting to know the extent to which staff have received education or training specifically on dementia? Perhaps some of these issues could be considered in the Discussion.

Small correction: note spelling ‘SEISTA’ on line 432 (Funding Sources)

Reviewer #3: Manuscript review: Describing and overcoming barriers to home staff using non-pharmacological strategies for sleep disturbances in care home residents with dementia: the SIESTA qualitative study. Submitted to PLOS ONE. December 2021

Thank you for the opportunity to review this paper. According to the title, the focus is on (1) “describing and overcoming barriers” to the use of non-pharmacological strategies for sleep disturbances for (2) people with dementia in residential care. I comment on these two issues as they do not seem to be the focus of the paper. Information on “barriers to managing sleep disturbances” (Results) is not presented until page 12 (line 259) and appears minimal compared to the presented information on strategies. Further, on page 19 (line 414+) the authors write that “we did not check the diagnoses of the residents” so there is an assumption about residents having dementia that may be incorrect.

Financial disclosure – please correct the spelling/typographical error.

Page 1 (line 3): Suggest deleting the second mention of “care home”

Abstract (page 2)

Background – please correct the grammar in the initial sentences: “care homes” should be singular; “It is “should be “They are” (relating to sleep disturbances”

Methods: Clarify how many times you interviewed the staff.

Results: Information from “During the daytime ….” to “They used telecare to monitor night-time risk” seems to fit better in Methods. Given the title of the paper, I would expect data-based information about barriers to be presented in the Results.

Conclusions: The “range of techniques” mentioned here seems to fit better in Results. Line 40: “residents night-time needs” should be residents’ night-time needs

Page 3

Line 47: delete “base”

Line 48: Move “of hypnotics” to follow “randomised controlled trials; explain hypnotics

Line 55: Delete “however”

Line 59: Spell out NICE before you use the abbreviation

Line 62: Include further details about the evidence supporting multi-component approaches to sleep disturbances. Later, in your Results section, you talk about the strategies staff used. I need to understand these strategies and that understanding needs to come from the detailed information you present in your Introduction.

Line 64: End the sentence after “residents with dementia.” Next sentence could be “These factors include noise (17), light (18), other residents’ (correct the position of the apostrophe; and make this correction on page 4, line 68; and on page 15, line 318) wakefulness ….

Page 4

Line 75: End the sentence after “residents with dementia.” Delete mention of the interviews. Including this information suggests you have already published the results of the interviews in reference 3. If this is true, the current paper does not need to be published. Make sure your research question for the current paper is clearly stated.

Page 5

Line 93: Change “do” to “did”

Line 100: Include information on the length of the interviews and when they were completed

Line 104: Provide further details on “altered the guide iteratively”

Page 6

Line 110: Who transcribed the interviews? Need further details here. If the first author transcribed all the interviews, there is the risk of bias. You mention further down that two authors developed the thematic coding framework but I am confused about the sequence of events. Please explain what occurred when there was uncertainly or disagreement about coding.

Page 7

Line 122: Much of what is presented here seems to fit better in your earlier Methods section.

Line 132: Please describe your participants as well as refer to Table 1. I would like to know more about whether/how your 8+6 care staff were educated about evidence-based strategies.

Line 138: You need a transition here e.g., “Details are presented in the following sections” so that a reader knows details are forthcoming.

Line 139: Your subheading is “Daytime strategies” so on line 140, suggest deleting “To prevent night-time sleep disturbances”.

Page 9

Line 172: Change ‘do” to “go”

Page 15 - Discussion.

Line 308: Please address your research question - which, given your title, appears to be on addressing barriers – and explore the reasons for these. I feel that when you mention barriers, you are listing them and making general statements rather than delving into why they occurred and citing evidence to support how they might be overcome. I understand that your focus is on non-pharmacological strategies, but I think you need to address the drugs that residents have been prescribed. This is missing from the current paper and is relevant, particularly in relating to residents’ apparent fatigue and sleepiness during the day.

6. PLOS authors have the option to publish the peer review history of their article (what does this mean?). If published, this will include your full peer review and any attached files.

Reviewer #1: **Yes: **Aurel Popa-Wagner

Reviewer #2: No

Reviewer #3: No

---

## [Author Response · Author response to Decision Letter 0]

29 May 2022

Reviewers comments 

Reviewer #1: Staff used a range of techniques that often worked in reducing or preventing sleep

disturbance. Some common techniques, such as caffeinated drinks, may be

counterproductive. Non-pharmacological interventions should consider practical

difficulties in limiting daytime napping, identifying residents night-time needs; day-night

disorientation, and insufficient night staffing. This is an useful study. I find appropriate a short introduction in aging and sheep disturbances, useful.

• Thank you for your positive feedback. 

Reviewer #2: This is an interesting manuscript that reports a qualitative study on factors that affect sleep behaviour for people with dementia in residential care settings. An adequate number of interviews were undertaken across four facilities, and the results are presented in a well-structured form. Overall, an important contribution.

I would consider the manuscript to be very well constructed and have no particular suggestions regarding the methodology and results. However, the paper does bring to light certain practises and attitudes that perhaps could be further pursued or at least discussed. In such a qualitative study, it would have been useful to understand the care ‘lens’ through which aged care workers and nurses undertook their roles. While it is laudable that staff would pursue non-pharmacological means to reduce sleep disturbances, from the descriptions of attitudes, it was not clear how person-centred these approaches may be. While intentions may have been good, many of the described practises could be perceived as manipulative and coercive, especially around activities and attempts to promote wakefulness during the day. I am wondering if the authors may have data on attitudes regarding the autonomy and ability for residents to make choices about their daily and nightly activities? From some descriptions, certain practices could be interpreted as physical restraint. It may not be available, but it would be interesting to know the extent to which staff have received education or training specifically on dementia? Perhaps some of these issues could be considered in the Discussion.

• Thank you for the positive feedback. We unfortunately do not have data on attitudes regarding the autonomy and ability for residents to make choices about their daily and nightly activities. We have now discussed this in the discussion section including “Furthermore, some of the strategies staff use to promote wakefulness in residents in the daytime could also be perceived as potentially coercive, for example residents have preferences if they want to participate in activities or nap instead, but there is often discrepancies between care home routines and resident’s preferences (39).”

Small correction: note spelling ‘SEISTA’ on line 432 (Funding Sources)

• Thank you for noticing – now corrected. 

Reviewer #3: Manuscript review: Describing and overcoming barriers to home staff using non-pharmacological strategies for sleep disturbances in care home residents with dementia: the SIESTA qualitative study. Submitted to PLOS ONE. December 2021

Thank you for the opportunity to review this paper. According to the title, the focus is on (1) “describing and overcoming barriers” to the use of non-pharmacological strategies for sleep disturbances for (2) people with dementia in residential care. I comment on these two issues as they do not seem to be the focus of the paper. Information on “barriers to managing sleep disturbances” (Results) is not presented until page 12 (line 259) and appears minimal compared to the presented information on strategies. Further, on page 19 (line 414+) the authors write that “we did not check the diagnoses of the residents” so there is an assumption about residents having dementia that may be incorrect.

• Thank you for reviewing the paper, and in line with your feedback we have changed the title of the paper slightly to “How do care home staff use non-pharmacological strategies to manage sleep disturbances in residents with dementia: the SIESTA qualitative study” to reflect that barrier to managing sleep are a part of the paper, but not the whole focus. 

• And yes, though it is correct we did not check the diagnoses of the residents staff spoke about, however we have acknowledged that this could be both a limitation as they may not have had dementia, but also a strength as many people with dementia living in care homes will not have a recorded clinical diagnosis of dementia. Furthermore, the majority of staff we spoke to had worked in care homes for many years, some up to 25 years, and therefore we trusted their judgement in knowing which residents did and did not have dementia. 

Financial disclosure – please correct the spelling/typographical error.

• This has been done, thank you. 

Page 1 (line 3): Suggest deleting the second mention of “care home”

• We have now removed this in line with the title change. 

Abstract (page 2)

Background – please correct the grammar in the initial sentences: “care homes” should be singular; “It is “should be “They are” (relating to sleep disturbances”

• This has now been changed 

Methods: Clarify how many times you interviewed the staff.

• This has now been added 

Results: Information from “During the daytime ….” to “They used telecare to monitor night-time risk” seems to fit better in Methods. 

• We disagree as this is a description of the results. 

Given the title of the paper, I would expect data-based information about barriers to be presented in the Results. 

• We have now changed the title of the paper to How do care home staff use non-pharmacological strategies to manage sleep disturbances in residents with dementia: the SIESTA qualitative study. And we do also mention barriers in the results in terms of “Staff found minimising daytime napping difficult, described insufficient staffing at night to attend to reorient and guide awake residents and said residents frequently did not know it was night-time.”

Conclusions: The “range of techniques” mentioned here seems to fit better in Results. Line 40: “residents night-time needs” should be residents’ night-time needs

• We have now moved that sentence. We have added in the apostrophe. 

Page 3

Line 47: delete “base” 

• This has been done. 

Line 48: Move “of hypnotics” to follow “randomised controlled trials; explain hypnotics

• This has been moved and hypnotics and have been explained. “Hypnotic medications are a class of medication commonly used to manage sleep disturbances, and in people with dementia have with frequent adverse effects such as fractures, particularly hip fractures, and higher mortality (8).”

Line 55: Delete “however”

• This has been done. 

Line 59: Spell out NICE before you use the abbreviation

• The NICE acronym has been spelt out in the previous paragraph on line 53 already. 

Line 62: Include further details about the evidence supporting multi-component approaches to sleep disturbances. Later, in your Results section, you talk about the strategies staff used. I need to understand these strategies and that understanding needs to come from the detailed information you present in your Introduction. 

• We have already included details on what multi-component approaches are and recent pilot studies that have showed preliminary evidence for using them “The most recent NICE guidelines recommend personalised multi-component approaches to sleep disturbances, which include sleep hygiene, increasing exposure to daylight, and increasing exercise and personalised activities (12). Recent pilot studies provide preliminary evidence of the efficacy of multi-component interventions in community settings (12, 14-16).”

Line 64: End the sentence after “residents with dementia.” Next sentence could be “These factors include noise (17), light (18), other residents’ (correct the position of the apostrophe; and make this correction on page 4, line 68; and on page 15, line 318) wakefulness …. 

• This has now been corrected, thank you.

Page 4

Line 75: End the sentence after “residents with dementia.” Delete mention of the interviews. Including this information suggests you have already published the results of the interviews in reference 3. If this is true, the current paper does not need to be published. Make sure your research question for the current paper is clearly stated.

• We have now edited this section to say “Here we report results from the SIESTA (Sleep problems In dEmentia: interviews with care home STAff) study, where we explored the views and experiences of care home staff members in one-to-one semi-structured interviews on how sleep disturbances in residents with dementia are managed non-pharmacologically in practice and any barriers to implementing these strategies.” 

• We have now also moved the sentence with reference 3 to in the methods, on line 88. 

However, we have made it clear that other results from this study have already been published, and that this is a different paper focussing on different results of the study. 

Page 5

Line 93: Change “do” to “did” 

• This has been changed. 

Line 100: Include information on the length of the interviews and when they were completed

• We have now added that “Interviews lasted for 20-60 minutes, with an average of 40 minutes.” We have also now added that “Fourteen staff members were interviewed during a day shift, two were interviewed whilst working night shifts, and the last two were interviewed before and after working a night shift, respectively.”

Line 104: Provide further details on “altered the guide iteratively”

• We have now added that we “altered the guide iteratively, as and when new themes emerged in the interviews.”

Page 6

Line 110: Who transcribed the interviews? Need further details here. If the first author transcribed all the interviews, there is the risk of bias. You mention further down that two authors developed the thematic coding framework but I am confused about the sequence of events. Please explain what occurred when there was uncertainly or disagreement about coding.

• We have now added on page 6 that interviews were transcribed by an external transcription service. We have also further explained and expanded upon the coding sequence of events and what happened for coding disagreements: 

“We developed a thematic coding framework, from the first five interviews.. The data from the first five interviews were initially independently coded by LW and KP, who identified the main themes and then developed an initial coding framework in line with the study objectives. LW and KP then met and discussed and edited this framework togther. The initial coding framework was then discussed and agreed upon by all study authors. Each additional interview was coded independently by LW and KP using the framework, which was added to iteratively. and a They met after every 3-4 interviews were coded, and discussed any coding disagreements and a consensus was reached. If needed a third person (GL) was consulted for over any coding disagreements where a consensus was not reached.”

Page 7

Line 122: Much of what is presented here seems to fit better in your earlier Methods section.

• We have now moved some of this section into the method section. 

Line 132: Please describe your participants as well as refer to Table 1. I would like to know more about whether/how your 8+6 care staff were educated about evidence-based strategies.

• Unfortunately we do not have any information on whether or how staff were educated about evidence-based strategies in the homes they worked in. We have now described them further: 

“The staff interviewed were mostly female (demographics in Table 1), with five working five worked in a residential care homes, and 13 working in care homes providing nursing and residential care. Staff varied in terms of their ethnicity and their roles within the care homes, and their age ranged from 21-64 years. They had worked in care homes for between 1-25 years, with a median of 6 years of care home experience.”

Line 138: You need a transition here e.g., “Details are presented in the following sections” so that a reader knows details are forthcoming.

• Thank you. This has now been added to line 138. 

Line 139: Your subheading is “Daytime strategies” so on line 140, suggest deleting “To prevent night-time sleep disturbances”. 

• We have kept to “prevent night-time sleep disturbances” in as here we are talking about using strategies in the daytime they may help to keep residents awake and busy, and to make sure they eat, and help them to subsequently sleep well at night because they have napped less, been more active and are less hungry. 

Page 9

Line 172: Change ‘do” to “go”

• This has now been changed, thank you. 

Page 15 - Discussion. 

Line 308: Please address your research question - which, given your title, appears to be on addressing barriers – and explore the reasons for these. I feel that when you mention barriers, you are listing them and making general statements rather than delving into why they occurred and citing evidence to support how they might be overcome. I understand that your focus is on non-pharmacological strategies, but I think you need to address the drugs that residents have been prescribed. This is missing from the current paper and is relevant, particularly in relating to residents’ apparent fatigue and sleepiness during the day.

• We have changed the title slightly so there isn’t a mention of addressing barriers in it now as we agree that barriers are part of the results but not all. “How do care home staff use non-pharmacological strategies to manage for sleep disturbances in residents with dementia: the SIESTA qualitative study”. We have discussed barriers in the discussion, however we do not feel there is much evidence to cite which explains how these barriers can be overcome, and instead we have cited discussion points that we feel are relevant here. 

We have now discussed sleep medications in relation to residents’ sleepiness in the day – “Furthermore, the hypnotic medications residents are given for their sleep disturbances could also make them fatigued and sleepy in the day (37), leading to them needing to nap excessively during the daytime and possibly exacerbating residents’ sleep disturbances (3).”

---

## [Decision Letter · Decision Letter 1]

4 Jul 2022

PONE-D-21-30400R1How do care home staff use non-pharmacological strategies to manage sleep disturbances in residents with dementia: the SIESTA qualitative studyPLOS ONE

Dear Dr. Webster,

Thank you for submitting your manuscript to PLOS ONE. After careful consideration, we feel that it has merit but does not fully meet PLOS ONE’s publication criteria as it currently stands. Therefore, we invite you to submit a revised version of the manuscript that addresses the points raised during the review process.

The reviewers are overall satisfied with the revision that improved the overall quality of the study-presentation. One reviewer still has some minor suggestions that I kindly ask you to consider before I can make a final decision on publication. 

We look forward to receiving your revised manuscript.

Kind regards,

Sara Rubinelli

Academic Editor

PLOS ONE

Journal Requirements:

Reviewers' comments:

Reviewer's Responses to Questions

**Comments to the Author**

1. If the authors have adequately addressed your comments raised in a previous round of review and you feel that this manuscript is now acceptable for publication, you may indicate that here to bypass the “Comments to the Author” section, enter your conflict of interest statement in the “Confidential to Editor” section, and submit your "Accept" recommendation.

Reviewer #1: All comments have been addressed

Reviewer #2: All comments have been addressed

Reviewer #3: All comments have been addressed

2. Is the manuscript technically sound, and do the data support the conclusions?

Reviewer #1: Yes

Reviewer #2: Yes

Reviewer #3: Yes

3. Has the statistical analysis been performed appropriately and rigorously? 

Reviewer #1: Yes

Reviewer #2: Yes

Reviewer #3: N/A

4. Have the authors made all data underlying the findings in their manuscript fully available?

Reviewer #1: No

Reviewer #2: Yes

Reviewer #3: Yes

5. Is the manuscript presented in an intelligible fashion and written in standard English?

Reviewer #1: Yes

Reviewer #2: Yes

Reviewer #3: Yes

6. Review Comments to the Author

Reviewer #1: The authors have adequately addressed my concerns. Good study albeit not being very innovative. The manuscript can be published in its present form

Reviewer #2: A thoughtful revision - my concerns have been largely addressed. The Paper will be of interest in the care sector.

Reviewer #3: Thank you for your careful attention to my earlier comments and suggestions.

I have a few more for clarity and grammar:

- Please check your use of "residents" and make the needed correction when you mean the possessive form, i.e. residents'

- On page 10, line 193: please change the heading to "Evening strategies to promote sleep" This then matches the earlier heading of "Daytime strategies..." I ask you to delete "good" as you have not measured the quality of sleep.

-On page 11, line 210: please change the heading to "Night-time strategies to manage sleep".

- Page 15, line 296: Change "these" to "sleep"

-Page 20, line 409: Change "is" to "are" (but there are often...)

- Page 21, line 431: Change "females" to "female"

Thank you

7. PLOS authors have the option to publish the peer review history of their article (what does this mean?). If published, this will include your full peer review and any attached files.

Reviewer #1: **Yes: **Aurel Popa-Wagner

Reviewer #2: **Yes: **James Vickers

Reviewer #3: No

---

## [Author Response · Author response to Decision Letter 1]

6 Jul 2022

Editors comments: 

We have now reviewed the reference list and found that no references were redacted and have sure references are as complete as possible. We updated the reference for a Cochrane review (McCleery, 2016) to a more recently updated version of this review (McCleery, 2020). We have also added in a few additional references in the data analysis section, and the strengths and weaknesses section. 

Reviewers' comments:

4. Have the authors made all data underlying the findings in their manuscript fully available?

Reviewer #1: No

Reviewer #2: Yes

Reviewer #3: Yes

All data have been made available and this was specified in the response to reviewers after the first review. 

6. Review Comments to the Author

Reviewer #1: The authors have adequately addressed my concerns. Good study albeit not being very innovative. The manuscript can be published in its present form

Reviewer #2: A thoughtful revision - my concerns have been largely addressed. The Paper will be of interest in the care sector.

Reviewer #3: Thank you for your careful attention to my earlier comments and suggestions.

I have a few more for clarity and grammar:

- Please check your use of "residents" and make the needed correction when you mean the possessive form, i.e. residents'

This has now been checked & edited where necessary. 

- On page 10, line 193: please change the heading to "Evening strategies to promote sleep" This then matches the earlier heading of "Daytime strategies..." I ask you to delete "good" as you have not measured the quality of sleep.

This has now been changed.

-On page 11, line 210: please change the heading to "Night-time strategies to manage sleep".

This has now been changed.

- Page 15, line 296: Change "these" to "sleep"

This has now been changed.

-Page 20, line 409: Change "is" to "are" (but there are often...)

This has now been changed.

- Page 21, line 431: Change "females" to "female"

This has now been changed.

---

## [Editor Report · Decision Letter 2]

27 Jul 2022

How do care home staff use non-pharmacological strategies to manage sleep disturbances in residents with dementia: the SIESTA qualitative study

PONE-D-21-30400R2

Dear Dr. Webster,

We’re pleased to inform you that your manuscript has been judged scientifically suitable for publication and will be formally accepted for publication once it meets all outstanding technical requirements.

Kind regards,

Sara Rubinelli

Academic Editor

PLOS ONE
---

## [Editor Report · Acceptance letter]

1 Aug 2022

PONE-D-21-30400R2 

How do care home staff use non-pharmacological strategies to manage sleep disturbances in residents with dementia: the SIESTA qualitative study 

Dear Dr. Webster:

I'm pleased to inform you that your manuscript has been deemed suitable for publication in PLOS ONE. Congratulations! Your manuscript is now with our production department. 

Kind regards, 

on behalf of

Dr. Sara Rubinelli 

Academic Editor

PLOS ONE